# Do Generative Models Learn Rare Generative Factors?

## Abstract

Generative models are becoming a promising tool in AI alongside discriminative learning. Several models have been proposed to learn in an unsupervised fashion the corresponding generative factors, namely the latent variables critical for capturing the full spectrum of data variability. Diffusion Models (DMs), Generative Adversarial Networks (GANs) and Variational Autoencoders (VAEs) are of particular interest due to their impressive ability to generate highly realistic data. Through a systematic empirical study, this paper delves into the intricate challenge of how DMs, GANs and VAEs internalize and replicate *rare* generative factors. Our findings reveal a pronounced tendency towards memorization of these factors. We study the reasons for this memorization and demonstrate that strategies such as spectral decoupling can mitigate this issue to a certain extent[1]

## 1 Introduction

In recent years, the machine learning field has witnessed a significant increase in the popularity and advancement of generative models (Scao et al., 2022; OpenAI, 2022; Taylor et al., 2022; Zhang et al., 2022b; Iyer et al., 2022; Touvron et al., 2023). These models have significantly advanced approaches to e.g. image generation and natural language processing, demonstrating the ability to create outputs that closely resemble real-world data (e.g. Karras et al. (2020); Zhang et al. (2022a)). The ongoing development and increasing adoption of these technologies, particularly large language models, have garnered substantial attention from academia and industry, while also becoming a topic of public interest (De Angelis et al., 2023; Mohamadi et al., 2023).

At the heart of these generative models lies the concept of *generative factors* (also known as factors of variation, or latent variables), which fundamentally affect the characteristics of the generated outputs (Liu et al., 2023; Bengio et al., 2013; Higgins et al., 2018; Träuble et al., 2021). These factors encompass many elements, from simple attributes such as colour or size in images to more complex features like sentence structure or thematic elements in text. Understanding and manipulating these generative factors is a key to harnessing the full potential of generative models (Fard et al., 2023; Yang et al., 2021; Shao et al., 2017).

Despite extensive research surrounding generative models (Bond-Taylor et al., 2022), one aspect remains notably under-explored: their ability to learn and replicate ***rare** generative factors*. Rare generative factors (RGFs) are latent variables which are highly skewed in their frequency of appearance in the real world (and hence in datasets) but play a critical role in the underlying data generating process. RGFs appear across a wide array of applications, including medical imaging (Liu et al., 2022), natural language generation (Mercatali & Freitas, 2021), and others.

**A motivating example** Consider a dataset composed of electrocardiogram (ECG) recordings with the RGF being the presence of the Brugada Syndrome, a rare disorder that can lead to sudden cardiac arrest. This syndrome is more prevalent in people in their 30s or 40s (Speranzon et al., 2024) but can also occur in childhood (Peltenburg et al., 2022). A dataset collected of patients having the disease is hence more likely to have individuals aged 30 to 50 with the disease. Generative models could generate new data to enrich dataset diversity, enhancing AI-based diagnostic tools or facilitating the early detection of this syndrome across a wider patient population, ultimately leading to timely interventions and more precise medical prognoses.

---

[1]The code will be made available upon acceptance.

This goal requires that generative models not only replicating the distinct ECG patterns associated with the syndrome within the subset of recordings where it is predominantly found, but also introducing these patterns into ECG recordings across other ages not commonly associated with the syndrome.

Focusing on Generative Adversarial Networks (GANs), Variational Autoencoders (VAEs) and Diffusion Models (DMs), in this paper we take a step forward by exploring their ability to capture these rare generative factors. We introduce a framework specifically designed to examine the effect of rarity in generative factors on the learning process of generative models. Focusing on simple canonical models (i.e. the original (plain) GAN architecture (Goodfellow et al., 2014), the standard VAE, a simple Denoising Diffusion Probabilistic Models (Ho et al., 2020)) allows us to distill insights without the confounding effects of additional complexities introduced in variant models, maintaining focus on core learning dynamics across all three model types.

By taking rarity to the extreme, considering datasets where the skew in the distribution of generative factors is pronounced, we pose a fundamental question: *When faced with a dataset that is heavily skewed in terms of the coverage of the generative factors, will a generative model successfully learn rare generative factors?* Addressing this question is crucial to understanding the limits of current generative models and developing new methodologies that can better capture and represent the diversity of generative factors, especially those that are rare. This exploration not only aims to enhance the fidelity and diversity of model-generated outputs but also seeks to contribute to the broader discourse on model robustness and fairness when dealing with skewed data distributions.

We show that plain GAN, VAE, and DM generally struggle to learn RGFs, tending instead to *memorize* them. This memorization is distinct from the memorization of individual training examples, as highlighted by recent studies. For instance, de Wynter et al. (2023) demonstrated how large language models exhibit example memorization, while Carlini et al. (2023) found that diffusion models tend to reproduce training examples during test time. Maini et al. (2023) showed that example memorization can be distributed across various neurons and layers, and Akbar et al. (2023) demonstrated memorization in diffusion models for synthetic brain tumour images. However, to the best of our knowledge, the memorization of generative factors remains significantly under-explored in the literature of generative models (Jegorova et al., 2023).

Our work provides valuable insights into the limitations of current generative models in learning robust, transferable representations from imbalanced datasets, opening new avenues for improving their generalization capabilities.

To summarise, we make three main **contributions**:

- A framework designed to systematically study the learning of RGFs in generative models.

- A statistical testing pipeline using z-scores and p-values to quantify the extent of memorization and assess factor-wise generalization at a class-specific level, rather than relying on global distribution metrics.

- A baseline comparison using matched datasets (balanced vs. skewed) to control for confounding variables and isolate the impact of data skew on generative learning performance.

- Through an extensive empirical study, we evaluate the capability of GANs, VAEs and DMs to learn and replicate RGFs, providing valuable insights into the dynamics of generative learning in the presence of data rarity.

- We identify and discuss the limitations in the context of RGF learning, explore the underlying reasons for these limitations, and evaluate a potential mitigation strategy specifically for GANs.

## 2 Related Work

Generative models can replicate the data distribution they are trained on but this is *not* what we aim for. We focus on a crucial aspect of unsupervised feature extraction: the ability to disentangle and generalize

RGF. We deliberately create skewed datasets where specific generative factors are present only in one class, not to test if models can mimic this distribution, but to examine if they can abstract these factors. Hence we focus not on how well models reproduce training data statistics, but on their capacity to learn generalizable latent representations from biased inputs. The tendency of models to memorize rare factor-class associations, rather than extending them to other classes, reveals a limitation in their ability to discover the underlying data generating process (Liu et al., 2022). This memorization of generative factors, highlights a significant challenge in unsupervised representation learning. It underscores the difficulty these models face in separating class-specific features from generalizable attributes when presented with skewed data. We also differentiate our focus on RGFs from the causal disentanglement approaches highlighted by Zhang et al. (2024). While Zhang et al. (2024) provide identifiability guarantees for disentangling causal variables using soft interventions, their emphasis lies in leveraging interventions to establish robust causal structures. Our study takes a different path, examining how generative models manage RGFs under extreme data imbalance. Unlike causal disentanglement, we make no assumptions about causality or intervention-based data. Instead, we investigate the mechanisms behind the memorization and generalization of RGFs, shedding light on the strengths and limitations of generative models in representing underrepresented factors (i.e. RGFs). This perspective offers a complementary angle to the causal disentanglement literature, enriching the broader discourse on disentanglement in generative modeling.

Garrido et al. (2020) primarily emphasize the prediction of rare feature combinations in population synthesis (i.e. zero-cell problem), particularly for unique categorical features, using Variational Autoencoders (VAE) and Wasserstein Generative Adversarial Networks (WGAN). While this approach provides valuable insights into modeling rare occurrences (addressing zero-cell problem), it operates at a narrow granularity that often overlooks broader dependencies and emergent patterns spanning the full image (or in raw data). This limitation risks underestimating the inherent complexity of rare variations, particularly when these involve intricate feature interdependencies.

Garrido et al. (2020) evaluate the learning of unique feature combinations by focusing on sampling zeros (i.e., logically possible combinations absent from the training data but present in test data) and structured zeros (i.e., logically impossible absent from training and test data). However, this methodology confines rarity to isolated feature combinations, limiting the model's ability to capture the interplay of features across the full image context. Consequently, the proposed evaluation framework lacks a controlled experimental baseline (i.e. comparison of generated sampling zeros while trained on balanced and highly skewed data). Our problem differs from the zero-cell problem because it focuses on image generation, where rare-generated factors (RGFs) dynamically affect the dataset. Some RGFs, like colour, thickening, or thinning, influence the entire image globally, while others, like fractures or swelling, have random, localized effects on parts of the images. Unlike the zero-cell problem, which deals with fixed absences in specific feature combinations, our challenge involves ensuring models can learn and generalize these rare and dynamically applied factors across samples. Moreover, evaluating the zero-cell problem is straightforward when dealing with tabular data, as it involves directly comparing whether the generated samples match values present in the test data. However, this approach cannot be directly applied to image generation tasks, where the outputs are in raw form, high-dimensional, and lack discrete categories for direct comparison.

Our study addresses these gaps by implementing a controlled experimental setup and constructing datasets where rarity is defined at the level of the entire image. Additionally, an evaluation framework has also been proposed to address this issue systematically. This approach enables a more comprehensive assessment of how generative models, including VAEs, GANs, and Diffusion Models (DMs), handle rare factors (RGFs) present in raw data (i.e. images).

## 3 Preliminaries

Consider a dataset $\{(\mathbf{x}_i, f_i, y_i)\}_{i=1}^n$, where $\mathbf{x}_i \in \mathcal{X}$ is a data instance, $f_i \in \{0, 1\}$ is a binary[2] *generative factor* and $y_i \in \{1, ..., C\}$ is a class label. For example, $\mathbf{x}_i$ is an image of a digit, $f_i$ indicates the color (green for 0, red for 1), and $y_i$ is the value of the digit.

---

[2]Our work can be extended to non-binary generative factors.

Central to our work are the generative factors, informally defined as:

**Definition 1 (Generative Factors, informal)** *The generative factors are the underlying latent variables that fully characterise the variation of the data in the domain $\mathcal{X}$.*

Our work focuses on the case of rare generative factors, formally defined as follows:

**Definition 2 (Rare Generative Factor, RGF)** *For $c \in \{1, ..., C\}$, let $S_{c,0} = \{i | y_i = c \text{ and } f_i = 0\}$ and $S_{c,1} = \{i | y_i = c \text{ and } f_i = 1\}$. A generative factor $f$ is rare if there exists a class $k \in \{1, ..., C\}$ such that $|S_{k,0}| \ll |S_{k,1}|$ and for all $c \neq k$, $|S_{c,0}| \gg |S_{c,1}|$.*

Intuitively, a dataset with a RGF is skewed. In this paper, we take the skewness to the extreme[3] and consider the case where $|S_{k,0}| = 0$ for a particular class $k$ and $|S_{c,1}| = 0$ for all other classes $c \neq k$.

Definition 2 characterizes a rare RGF as one whose distribution is highly skewed with respect to the class label. Specifically, the factor $f$ is considered rare if, for some class $k \in \{1, ..., C\}$, it appears exclusively (or overwhelmingly) in class $k$, and is absent (or nearly absent) in all other classes. The sets $S_{c,0}$ and $S_{c,1}$ represent the indices of samples in class $c$ where the generative factor $f$ takes value 0 or 1, respectively. The condition $|S_{k,0}| \ll |S_{k,1}|$ and $|S_{c,0}| \gg |S_{c,1}|$ for all $c \neq k$ implies that $f = 1$ is strongly concentrated in class $k$. This definition implicitly captures a significant variation in the conditional distribution $\mathbb{P}(f \mid y)$ across classes. In the extreme case we study, this variation is taken to its limit: $f = 1$ occurs only in class $k$, and $f = 0$ in all other classes. This setup enables a controlled analysis of whether generative models generalize the factor $f$ beyond class $k$ or memorize its co-occurrence, thereby disentangling generalization from class-conditional memorization.

Note that we *only* use the data instances $\mathbf{x}_i$ for the training of generative models. Generative factors $f_i$ and class labels $y_i$ serve *exclusively* to evaluate (after training) the model's ability to learn the generative factors. This setting reflects real-world scenarios where explicit labels or factors might not be readily available, challenging the model to capture the generative factors accurately.

## 3.1 Examples

We now briefly discuss motivating real-world examples of rare generative factors. For each example, we provide a detailed description of the role of $\mathbf{x}_i$, $f_i$ and $y_i$.

**Example 1: Medical Imaging for Brain Health Across Different Ages**

- $\mathbf{x}_i$ - MRI scan of the brain.
- $f_i$ - A binary generative factor indicating the age group of the patient, either young (under 60) or old (60+).
- $y_i$ - The health condition identified by the scan, such as normal aging, mild cognitive impairment, or Alzheimer's disease.

In this example, the distribution of age is skewed because Alzheimer's disease mostly affects older people. Consequently, learning to understand the concept of age in relation to Alzheimer's and generating MRI images that accurately depict Alzheimer's in younger individuals, which is still possible with early-onset Alzheimer's (Mendez, 2019), poses a significant challenge. This difficulty arises from the rarity of early-onset Alzheimer's cases in younger populations, making it difficult for models to capture and replicate this condition accurately in generated images.

**Example 2: Text Style in Literary Genres**

- $\mathbf{x}_i$ - A passage of text.
- $f_i$ - A binary generative factor indicating the text style, e.g. whether the text includes archaic English words or not (a modern style).
- $y_i$ - The literary genre of the text, such as modern fiction, contemporary poetry, or historical fiction.

---

[3]We relax it in Appendix E.

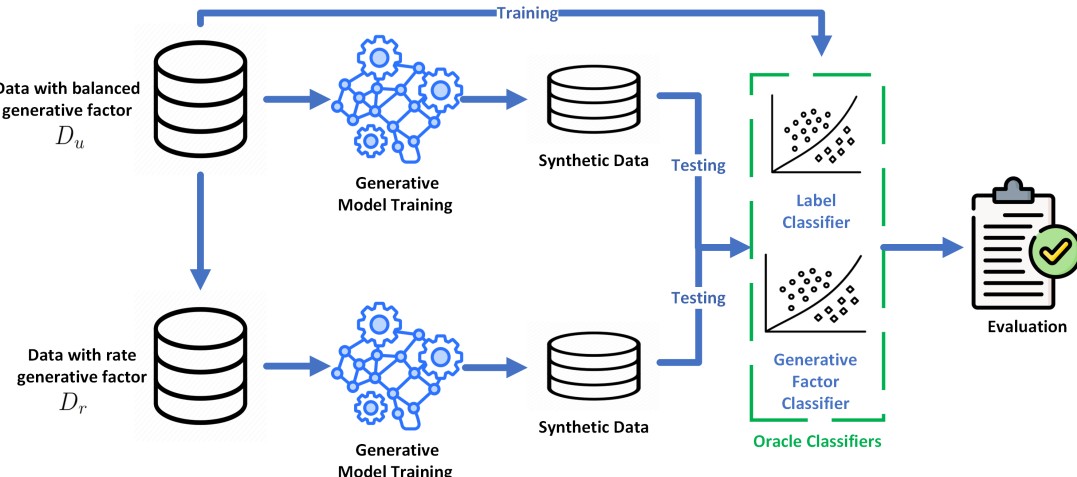

Figure 1: Framework for assessing the learnability of rare generative factors.

In this example, text style might be a rare generative factor, since archaic English is uncommon in modern fiction and contemporary poetry but frequently found in historical fiction. The challenge for generative models is to learn the concept of text style from such skewed data.

**Example 3: Car Images in Urban and Rural Environments**

- $\mathbf{x}_i$ - Image of a car.
- $f_i$ - The environment in which the car is captured, urban or rural.
- $y_i$ - The brand of the car.

In this example, the rarity of the generative factor arises because luxury car brands, such as BMW, are frequently observed in urban landscapes but are considerably less common in rural environments. This discrepancy presents a challenge in learning the generative factor of the environment effectively.

## 4 Framework for Assessing the Learnability of RGFs

We now present our framework for studying the learnability of RGFs, illustrated in Figure 1.

**Setup:** We start our investigation with a dataset $D_u = \{(\mathbf{x}_i^{(u)}, f_i^{(u)}, y_i^{(u)})\}$ characterized by a *uniform* distribution of the generative factor; that is, within each class, the number of samples with $f_i = 1$ equals those with $f_i = 0$. This balanced dataset serves as a baseline for understanding how generative models perform under standard conditions, where no generative factor is particularly rare.

To understand the impact of an RGF, we construct a new dataset, $D_r = \{(\mathbf{x}_i^{(r)}, f_i^{(r)}, y_i^{(r)})\}$, derived from the original data instances in $D_u$. In this tailored dataset, we introduce a *deliberate* skew: for some selected class $k$, all examples have $f_i = 1$, which signifies the presence of the RGF. In contrast, for all other classes $c \neq k$, all examples have $f_i = 0$, indicating the absence of this factor. These two datasets ($D_u$ and $D_r$) allow us to closely examine how the presence of a rare generative factor influences the learning and generative capabilities of generative models.

To this end, we train two separate generative models (of the same type) for $\{\mathbf{x}_i^{(u)}\}$ and $\{\mathbf{x}_i^{(r)}\}$, respectively. From each trained model, we then generate $M$ samples for evaluation. To evaluate these generated samples, we employ two oracle classifiers. These classifiers are trained on the balanced dataset $D_u$, serving two functions:

1. **Label Classifier:** This classifier is trained using data pairs $\{(\mathbf{x}_i^{(u)}, y_i^{(u)})\}$, which consist of the data instances and their corresponding class labels. Its role is to categorize the generated samples

into the correct classes, assessing the model's ability to maintain class-specific characteristics in the generated data.

2. **Generative Factor Classifier:** This binary classifier, trained on $\{(\mathbf{x}_i^{(u)}, f_i^{(u)})\}$ pairs, focuses on identifying the presence or absence of the generative factor within each sample.

We ensure that both classifiers achieve high accuracy (on a separate test set).

Next, we use the classifiers to determine both the class label and the binary generative factor for each of the $M$ samples produced by the respective generative model, and then calculate the distribution of the generative factor for each class $c$. We denote by $P_c^{(u)}$ the proportion of instances with $f = 1$ within class $c$, generated by the generative model trained on the uniformly distributed dataset $D_u$. Similarly, $P_c^{(r)}$ represents the proportion of instances with $f = 1$ from class $c$, generated by the generative model that is trained on the skewed dataset $D_r$.

**Our hypothesis** We hypothesize that for each class $c$, the proportion of generated instances by both trained models will be comparable. This hypothesis is grounded in the notion that effective learning by generative models should allow them to extract the generative factors, regardless of their rarity in the training data, with a high degree of fidelity. Essentially, this suggests that the models' ability to discern and generate generative factors is *not* significantly hindered by the skewed distribution of these factors in the training dataset.

**Assessing the learning of RGF** We perform a statistical test of the hypothesis to compare the proportions $P_c^{(u)}$ and $P_c^{(r)}$. We employ a one-sample z-test, which allows us to determine whether the observed differences in proportions between the two groups are statistically significant. We denote by $z_c$ the z-score[4] corresponding to class $c$,

$$z_c = \left(P_c^{(r)} - P_c^{(u)}\right)/\sqrt{\frac{P_c^{(u)}\left(1 - P_c^{(u)}\right)}{M}} \ . \tag{1}$$

To evaluate the capability of generative models to learn RGFs, we calculate the p-value associated with each computed z-score $z_c$ for class $c$. When p-value $> 0.05$, we uphold the null hypothesis, which implies that the model has effectively *learned* the generative factor. This outcome suggests that there is no significant difference between the expected and observed frequencies of the RGF among the generated instances, indicating successful learning by the generative model.

Conversely, a p-value less than 0.05 leads to the rejection of the null hypothesis. Specifically, for the class $k$ where the rare generative factor has been introduced, and where $z_k > 0$, this outcome signifies that the generative model has not learned but rather *memorized* the generative factor for this class. Similarly, if we observe a p-value below 0.05 for a class $c \neq k$ accompanied by $z_c < 0$, this also indicates memorization of the generative factor by the generative model for classes other than $k$. It is noteworthy to mention that deviations from these specified conditions are rare in practice, underscoring the models' tendency to either learn or memorize generative factors. The subsequent section details the datasets and the specific generative factors employed in our study.

**Justification for the Chosen Test Statistic.** We employ a one-sample $z$-test to compare the class-wise proportions $P_c^{(u)}$ and $P_c^{(r)}$ of the generative factor in the synthetic data generated from balanced and skewed training datasets, respectively. The $z$-test is appropriate in our setting because we are comparing an observed proportion $P_c^{(r)}$ against a reference population proportion $P_c^{(u)}$ obtained from the balanced dataset. Given the sufficiently large number of generated samples ($M = 1000$), to ensure that the sampling distribution of the proportion approximates a normal distribution, satisfying the assumptions of the test. This test offers an interpretable and computationally efficient means of quantifying deviations from the expected behaviour under the null hypothesis that the model has learned the generative factor in a generalizable way. Moreover, the $z$-score provides not just significance testing but also directionality (i.e., whether the factor appears more or less frequently than expected), which is critical for distinguishing between learning and memorization. We selected the one-sample $z$-test due to its simplicity, suitability for proportion data in large samples.

---

[4]The $z$ notation should not be confused with a latent space.

# 5  Dataset and Generative Factors

In this work we primarily utilized the Colored-MNIST dataset (Arjovsky et al., 2020) and the Morpho-MNIST dataset (Castro et al., 2019), both are stylish versions of the classical greyscale handwritten digits classification MNIST dataset (LeCun et al., 1998). The Colored-MNIST dataset enhances the original digit images by incorporating a color scheme of green and red. The Morpho-MNIST dataset modifies the digits with morphological modifications, such as variations in thickness, swelling, and the introduction of fractures. To extend our analysis beyond handwritten digits, we also employed a subset of the Comprehensive Cars (CompCars) Surveillance dataset (Yang et al., 2015). From this dataset, we selected images of two car makes (Volkswagen and Toyota) in two colours (black and white), allowing us to explore our hypotheses in a different domain. Table B.2 in Appendix B details the sample distribution of our CompCars subset.

We designed our VAE, GAN and DM to work with RGB (3 channels) images. Consequently, to accommodate the greyscale images from the Morpho-MNIST dataset, we transformed them into colour images. This is achieved by randomly assigning either a red or a green colour to each image, ensuring an equal probability distribution between the two colours for the images with morphological modifications.

As detailed in Section 4, for each generative factor under consideration we created two datasets:

1. A balanced dataset $D_u$, where the generative factor is uniformly distributed across all classes. For MNIST-based experiments, this dataset comprises 60000 images with an equal representation of each digit. In the case of the CompCars subset, we utilized 1448 images, ensuring an even distribution between Volkswagen and Toyota cars.

2. A dataset $D_r$ with rare generative factor. For MNIST-derived datasets, we introduce the rare generative factor to a single digit class. We specifically chose digits "1" and "2" as representative cases, conducting separate experiments where the rare factor is exclusively associated with each of these digits. This approach allows us to examine how the shape of the digit might influence the model's ability to learn or memorize the rare factor. For the CompCars subset, we assign the rare generative factor to car make.

We trained VAE, GAN and DM separately on each dataset. The full training details and model architectures are described in Appendix A.

After training the models for each generative factor, we generated $M = 1000$ synthetic images. The oracle classifiers are used to detect the class (digit for MNIST, car make for CompCars) and the presence of the generative factor in the synthetic images.

## 5.1  Generative Factors

Variations in colour and morphology are naturally used in our work as generating factors, as they are important in determining the visual appearance of the digits. Specifically, we defined the following 5 generative factors for digits: Colour, Fracture, Thinning, Thickening, and Swelling. Note that *only* one generative factor is introduced at a time. Figure D.2 (see Appendix D) demonstrates the case of **rare** generative factors where digit "1" is selected as the class in which the generative factor is introduced (for example, for the Thickening factor all images of digit "1" are thick while other digits retain a standard thickness). For the colour factor, the presence of green is designated as the rare generative factor. For CompCars, colour is the generative factor, where all Volkswagen cars are white and Toyota cars are black.

For digits, the generative factors are introduced in the images using the Morpho-MNIST python library.[5] For Thinning and Thickening the value of the *amount* parameters is 0.7 and 1, respectively. For Swelling the value of the *strength* parameter is 3 and the *radius* is 7. For Fracture the value of *num_frac* is 3. For cars, the generative factor is introduced by selecting the corresponding subset of the CompCars dataset.

---

[5]https://github.com/dccastro/Morpho-MNIST

## 5.2 Oracle Classifiers

As mentioned in Section 4, we rely on oracle classifiers to categorize images generated by VAEs, GANs and DMs. We employed Convolutional Neural Networks (CNN) as our oracle classifiers. The details of the architectures appear in Appendix A. For each generative factor we trained two oracle classifiers on the balanced dataset. For the MNIST-derived datasets, we trained one classifier for digit classification and another for factor classification, resulting in a total of 10 classifiers. Some images from the dataset used to train the digit classifier (10-class problem) and colour classifier (2-class problem) appear in Figure B.1 (see Appendix B). For cars, we trained one classifier for car make classification and another for colour classification, using the data shown in Table B.2.

The MNIST oracle classifiers are trained using SGD for 8 epochs employing the cross entropy loss, batch size of 64, learning rate of 0.01, and momentum of 0.5. For car make classification, we used 100 epochs. To evaluate the performance of these classifiers, we used a test-set of 20000 samples for digits and 185 samples for cars. The classification accuracies, as detailed in Table B.1, show that all classifiers achieved a test-set accuracy exceeding 92%, underscoring their high efficacy in accurately identifying both digits, car make and generative factors.

# 6 Results and Discussion

Utilizing the framework of Section 4 and the datasets (Section 5), we now present our findings. Due to space constraints, we have placed the majority of tables and figures in the Appendix.

Initially, we used the balanced datasets $D_u$ for each RGF, trained the models, and then generated $M = 1000$ synthetic images. As expected, $P_c^{(u)}$ approximates 0.5 in the majority of cases, indicating a balanced representation of the generative factors within the synthetic images (for details see Tables C.3 and C.4 in Appendix C).

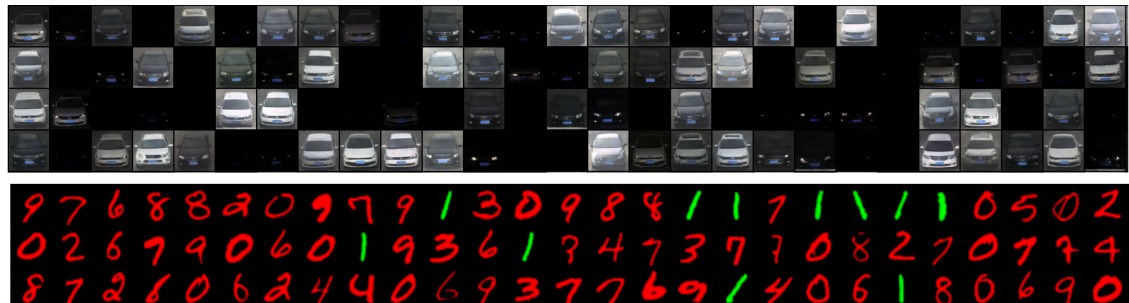

Figure 2: Some generated images by a Diffusion model trained on CompCars and Colored-MNIST skewed datasets.

Subsequently, for each RGF, we trained the models using the skewed dataset $D_r$ and determined the proportions $P_c^{(r)}$ for each digit (for MNIST dataset) and car (for CompCars dataset). We then used Eq. (1) to calculate the z-scores and report the results in Tables 1, 2 and 3.

While standard machine learning theory predicts that underrepresented features are difficult to learn due to their low empirical frequency, our focus on rare generative factors (RGFs) highlights an important distinction: RGFs actively shape the data generation process, meaning their absence or distortion can affect not only classification but also the model's ability to generate coherent, semantically consistent samples. This makes their impact more profound than that of merely rare labels or attributes. Theoretically, RGFs define a low-probability region in the generative manifold, and standard likelihood-based training may fail to sufficiently penalize errors in such regions. This underlines the need for tailored inductive biases or priors that preserve generative completeness, especially in applications where coverage of rare modes is critical.

Table 1: z-scores for all models (VAE, GAN without SD, GAN with SD, DM) where all images of digit "1" have RGF. Bold: similar proportions ($p > 0.05$), indicating RGF learning.

| Digit | Colour VAE/GAN/GAN-SD/DM | Fracture VAE/GAN/GAN-SD/DM | Swell VAE/GAN/GAN-SD/DM | Thick VAE/GAN/GAN-SD/DM | Thin VAE/GAN/GAN-SD/DM |
|---|---|---|---|---|---|
| 0 | - / - / - / - | **-1.80** / **0.01** / **1.36** / -28.56 | -5.28 / -4.77 / **0.89** / -6.51 | -4.66 / -9.09 / - / -9.28 | -3.70 / -8.65 / **0.82** / -40.68 |
| 1 | -6.14 / 17.05 / -5.49 / 14.77 | 3.92 / **-0.97** / **1.44** / 32.75 | **-0.94** / 4.23 / -5.68 / 9.57 | 2.39 / 2.96 / -4.97 / 26.33 | 7.15 / 22.90 / **0.16** / 14.54 |
| 2 | - / -40.92 / -2.34 / - | **-1.71** / -15.49 / -2.34 / -4.42 | -8.48 / -4.87 / **-0.29** / -9.43 | -7.11 / -4.36 / -7.89 / -12.10 | -2.36 / -5.82 / -7.17 / -16.81 |
| 3 | -24.94 / -37.48 / -82.23 / - | -2.30 / -10.30 / -5.54 / -14.90 | -2.19 / -5.89 / -11.27 / -6.85 | -12.21 / -8.36 / -4.25 / -14.93 | -3.62 / -19.58 / -7.74 / -50.81 |
| 4 | - / - / - / - | **0.03** / -15.20 / -5.08 / -37.92 | -7.23 / -14.59 / -9.91 / -7.60 | -5.97 / -56.40 / -16.55 / -8.45 | **-1.23** / -8.71 / -4.45 / -15.66 |
| 5 | - / - / - / - | **0.59** / -4.26 / -2.48 / -11.92 | -3.55 / -9.86 / -16.00 / -9.21 | -22.98 / -19.24 / -15.89 / -20.45 | -3.60 / -12.31 / -4.39 / -12.13 |
| 6 | - / -34.87 / -4.93 / - | **-1.65** / -34.87 / -4.93 / -16.97 | -3.07 / -13.66 / -8.55 / -5.63 | -12.03 / -42.80 / -14.31 / -14.76 | -5.57 / -11.97 / -11.03 / -66.40 |
| 7 | - / -40.20 / -7.77 / - | **-0.79** / -16.46 / -7.77 / -14.11 | -10.78 / -7.93 / -9.53 / -13.31 | -2.38 / -8.80 / -6.09 / -22.88 | **-0.78** / -7.90 / **0.47** / -7.90 |
| 8 | -10.29 / -65.37 / -2.97 / - | -2.25 / **-0.87** / -2.97 / -14.22 | -5.66 / -8.03 / **-0.64** / -7.26 | **-1.34** / -14.22 / -3.38 / -23.75 | -5.59 / -11.85 / -11.35 / -13.32 |
| 9 | - / -11.09 / -6.50 / - | -5.48 / -11.09 / -6.50 / -14.44 | -8.57 / -12.33 / -3.48 / -7.04 | **-1.62** / -23.56 / -11.47 / -15.23 | **-1.25** / -11.60 / -7.83 / -6.49 |
| Total | -75.30 / -39.18 / -44.87 / -42.67 | -2.21 / -21.28 / -9.57 / -18.87 | -14.60 / -21.13 / -15.49 / -17.49 | -14.01 / -33.41 / -24.97 / -20.64 | -7.86 / -21.09 / -13.27 / -35.08 |

## 6.1 Memorization of RGF

Comparing the proportions $P_c^{(u)}$ and $P_c^{(r)}$ via the z-scores in Tables 1, 2 and 3 underscores the propensity of generative models to memorize RGFs. For instance, GAN exhibits a notable bias towards associating the green colour with digits "1" and "2", in contrast to the red colour, which is more frequently linked with the remaining digits. Specifically, when the green color is assigned to digit "1", an overwhelming 87% of generated images display this characteristic, a stark contrast to the 35% for the balanced data. Conversely, the presence of green in images of other digits is minimal, hovering around 1%, indicating a clear memorization of the green color for digit "1" without extending this rare factor to other digits. A similar trend is evident when the colour factor is applied to digit "2" (see Appendix D for detailed results).

The large z-scores highlight the significant differences in proportions between $P_c^{(u)}$ and $P_c^{(r)}$, confirming the memorization effect. This memorization phenomenon is *not* limited to colour in digit datasets. It extends, yet to varying degrees, across other generative factors we studied. In the case of car images, we observe a similar trend where the models tend to strongly associate colour with a car make. The observed pattern suggests a broader trend: *GANs and DMs exhibit a stronger tendency towards memorization of RGFs compared to VAEs*, both in digit recognition and car classification tasks. Visual inspection suggests that DM provides the highest image quality, as shown in Figure 2, but at the cost of increased memorization (the images generated using VAE and GAN are shown in Appendix D). This different behaviour across model types and datasets highlights the nuanced ways in which various generative architectures approach the challenge of learning from skewed data distributions.

**Distinguishing Memorization from Semantic Correlation.** A crucial distinction in our study is between (a) *memorization* of rare generative factors (RGFs) and (b) genuine *learning* of generative factors that are strongly correlated with semantic class features. Memorization, in our context, refers to the model reproducing RGFs only within the class where they were seen during training (e.g., generating green digits exclusively for class "1" if green was only present in that class). This indicates that the model has not abstracted the RGF as a transferable concept, but instead has tightly coupled it with the class identity. In contrast, learning is evidenced when the model applies the RGF to other classes not seen with that factor in training, thereby indicating that it has captured the generative factor independently of class label. To empirically distinguish the two, we rely on the distributional comparison between $P_c^{(r)}$ and $P_c^{(u)}$ using the z-test, where $P_c^{(u)}$ acts as a reference distribution under balanced conditions. A non-significant z-score ($p > 0.05$) suggests that the model has generalized the RGF across classes, whereas a significant z-score in the direction of the skew (i.e., high $P_c^{(r)}$ for the class with the RGF and low for all others) indicates memorization. Thus, semantic correlation alone is not sufficient to explain this behavior unless it holds under balanced data, in which case $P_c^{(u)}$ would already show asymmetry. Our framework explicitly controls for such effects by comparing against the balanced baseline.

## 6.2 How RGF memorization originates in GANs?

We are interested in understanding how memorization of RGFs happens. We picked GANs for two main reasons: first, because they exhibited a stronger tendency to memorize RGFs in our experiments compared

Table 2: z-scores for all models (VAE, GAN without SD, GAN with SD, DM) where all images of digit "2" have RGF. Bold: similar proportions ($p > 0.05$), indicating RGF learning.

| Digit | Colour VAE / GAN / GAN-SD / DM | Fracture VAE / GAN / GAN-SD / DM | Swell VAE / GAN / GAN-SD / DM | Thick VAE / GAN / GAN-SD / DM | Thin VAE / GAN / GAN-SD / DM |
|---|---|---|---|---|---|
| 0 | -20.84 / - / -78.05 / - | **-1.16** / **0.59** / 2.73 / -22.58 | **-1.63** / -4.54 / 3.36 / -24.44 | -9.86 / -8.20 / -4.92 / -21.55 | -4.11 / -10.53 / **1.74** / -42.08 |
| 1 | -23.41 / -12.64 / -22.99 / -89.1 | **-0.38** / -42.82 / -38.40 / -26.36 | -8.76 / -11.04 / -14.67 / -14.38 | -7.24 / -28.73 / -45.43 / -15.81 | -6.84 / -14.10 / -5.68 / -20.25 |
| 2 | 17.24 / 13.64 / 3.12 / 42.09 | **1.88** / **0.42** / -2.38 / **1.83** | 3.27 / **-0.40** / **-1.17** / 8.23 | 6.16 / 9.99 / **0.06** / 11.74 | 5.04 / 7.25 / -3.14 / 15.93 |
| 3 | -26.85 / -25.03 / -30.88 / - | -4.10 / **-0.65** / -2.84 / -6.92 | -4.10 / -4.31 / -11.34 / -6.57 | -13.58 / -15.00 / -10.94 / -36.83 | -2.26 / -32.70 / -11.01 / -18.24 |
| 4 | -43.88 / - / - / - | **-0.27** / -29.01 / -6.32 / -9.89 | -6.16 / -2.21 / -10.69 / -7.06 | -5.12 / - / -62.51 / -8.78 | -3.65 / -12.04 / -12.73 / -10.14 |
| 5 | - / - / - / - | -4.36 / **-0.07** / -4.39 / -3.67 | -2.00 / -4.89 / -11.96 / -21.07 | -22.69 / -43.46 / -22.24 / - | -2.87 / -16.09 / -9.24 / -12.53 |
| 6 | - / -49.63 / -16.42 / - | **-0.76** / -19.33 / -16.32 / -10.92 | -2.17 / -6.03 / -5.50 / -7.05 | -9.70 / -27.05 / -21.60 / -11.03 | -5.34 / -17.38 / -6.17 / -30.48 |
| 7 | -17.70 / -35.28 / - / -70.75 | -2.25 / -16.87 / -7.84 / -7.93 | -17.03 / -4.31 / -5.56 / -10.39 | -7.93 / -12.31 / -22.25 / -13.44 | **-1.28** / -17.33 / **0.17** / -20.8 |
| 8 | -55.44 / -45.78 / -8.21 / -69.9 | **-0.30** / -2.86 / -2.12 / -7.11 | -7.87 / -8.50 / -4.17 / -7.7 | **-1.91** / **-1.93** / -9.05 / -22.24 | -5.03 / -17.35 / -6.72 / -18.66 |
| 9 | - / - / - / - | -3.49 / -23.71 / -11.12 / -9.14 | -7.85 / -7.26 / -10.43 / -8.23 | -2.80 / -32.17 / -29.76 / -10.42 | -4.98 / -14.81 / -5.90 / -10.6 |
| Total | -39.94 / -37.66 / -42.60 / -47.28 | -4.27 / -21.74 / -19.12 / -20.83 | -14.81 / -15.71 / -19.33 / -20.83 | -17.05 / -34.28 / -43.87 / -23.01 | -9.95 / -35.36 / -15.23 / -32.41 |

Table 3: CompCars, z-scores for all models (VAE, GAN without SD, GAN with SD, DM), with colour RGF: white Volkswagen, black Toyota. Bold: similar proportions ($p > 0.05$), indicating RGF learning.

| Make | VAE | | | GAN | | | GAN-SD | | | Diffusion Models | | |
|---|---|---|---|---|---|---|---|---|---|---|---|---|
| | Black | White | z | Black | White | z | Black | White | z | Black | White | z |
| Volkswagen | 161 | 397 | 13.11 | 153 | 425 | 12.28 | 132 | 350 | 10.64 | 153 | 204 | 9.98 |
| Toyota | 336 | 106 | -11.33 | 334 | 88 | -8.16 | 454 | 64 | -17.05 | 605 | 38 | -19.45 |
| Total | 497 | 503 | 2.09 | 487 | 513 | 4.62 | 586 | 414 | **-1.67** | 758 | 242 | -2.07 |

to VAEs, and second, because their architecture includes a discriminator that allows us to explore the role of adversarial training in potentially encouraging this memorization behaviour. Indeed, we analysed the discriminator loss during GAN training with respect to the "real label" using a separate balanced validation set of 2000 images of digits and 185 images of cars.

To do this, we computed the loss only for images where RGFs are applied ("1" and "2" for MNIST and Volkswagen for CompCars). We differentiate between images featuring RGFs and those without.

Figure 3 illustrates the discriminator loss for the colour factor in MNIST data, with RGF present in digit "1" (Appendix D presents results for other RGFs and digits). In this plot, solid lines depict the loss associated with images containing RGFs (i.e. green images), while dashed lines indicate the loss for images lacking RGFs (i.e. red images). A green horizontal dashed line represents the threshold loss at the discriminator's decision boundary between identifying images as real or fake, corresponding to a loss of $\log(2)$ when the discriminator output logit is 0.

When training the GAN with the balanced dataset $D_u$, there appears to be no significant discrepancy between the loss for images with RGF and those without, suggesting that the discriminator does not differentiate based on the presence of RGF. In other words, the discriminator is invariant to RGF. However, training on the skewed dataset $D_r$, we observe a gap between the losses for images with and without RGF. This indicates that despite all images being "real", the discriminator classifies images with and without RGFs differently, losing its invariance to RGFs. This differentiation likely stems from the spurious correlation between the digit and the RGF, reminiscent of the "gradient starvation" phenomenon identified by Pezeshki et al. (2021) in the context of discriminative learning. This phenomenon, where the model excessively focuses on dominant features at the expense of others, may explain the discriminator's skewed learning, underlining the complexity of addressing memorization of RGFs in GANs.

### 6.3 Mitigating memorization in GANs by Spectral Decoupling

Our next focus is to evaluate if the Spectral Decoupling (SD) technique, previously proposed by Pezeshki et al. (2021) to address the issue of gradient starvation, can also help in reducing the memorization of RGFs by GANs.

In the context of discriminative learning, SD augments the loss function with a regularization term $\frac{\lambda}{2}\|\hat{\mathbf{y}}\|^2$, where $\lambda$ is a regularization strength hyperparameter, and $\hat{\mathbf{y}}$ is the logits vector output by the model for a given input batch. This regularizer aims to restrain the magnitudes of logits, thereby preventing any single (and potentially spurious) feature from overpowering the model's output.

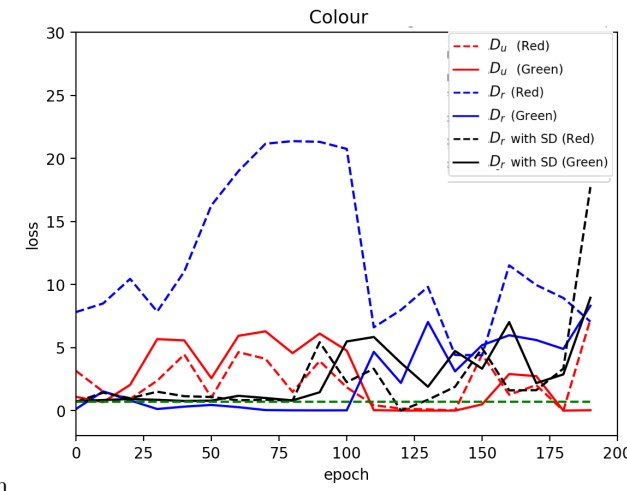

!h

Figure 3: Discriminator loss with respect to the "real label", where the colour RGF is introduced in digit "1".

Table 4: RGF learning (L) vs. memorization (M) summary. Notation: VAE/GAN/GAN-SD/DM. A total of 43 cases were learned out of 440.

| digit | RGF in digit 1 | | | | | RGF in digit 2 | | | | |
|---|---|---|---|---|---|---|---|---|---|---|
| | colour | frac | swell | thick | thin | colour | frac | swell | thick | thin |
| 0 | M/M/M/M | **L/L/L/M** | **M/M/L/M** | M/M/M/M | **M/M/L/M** | M/M/M/M | **L/L/M/M** | **L/M/M/M** | M/M/M/M | M/M/L/M |
| 1 | M/M/M/M | **M/L/L/M** | **L/M/M/M** | M/M/M/M | **M/M/L/M** | M/M/M/M | **L/M/M/M** | M/M/M/M | M/M/M/M | M/M/M/M |
| 2 | M/M/M/M | **L/M/M/M** | **M/M/L/M** | M/M/M/M | M/M/M/M | M/M/M/M | **L/L/M/L** | **M/L/L/M** | **M/M/L/M** | M/M/M/M |
| 3 | M/M/M/M | M/M/M/M | M/M/M/M | M/M/M/M | M/M/M/M | M/M/M/M | **M/L/M/M** | M/M/M/M | M/M/M/M | M/M/M/M |
| 4 | M/M/M/M | **L/M/M/M** | M/M/M/M | M/M/M/M | **L/M/M/M** | M/M/M/M | **L/M/M/M** | M/M/M/M | M/M/M/M | M/M/M/M |
| 5 | M/M/M/M | **L/M/M/M** | M/M/M/M | M/M/M/M | M/M/M/M | M/M/M/M | **M/L/M/M** | M/M/M/M | M/M/M/M | M/M/M/M |
| 6 | M/M/M/M | **L/M/M/M** | M/M/M/M | M/M/M/M | M/M/M/M | M/M/M/M | **L/M/M/M** | M/M/M/M | M/M/M/M | M/M/M/M |
| 7 | M/M/M/M | **L/M/M/M** | M/M/M/M | M/M/M/M | **L/M/L/M** | M/M/M/M | M/M/M/M | M/M/M/M | M/M/M/M | **M/M/L/M** |
| 8 | M/M/M/M | **M/L/M/M** | **M/M/L/M** | **L/M/M/M** | M/M/M/M | M/M/M/M | **L/M/M/M** | M/M/M/M | **M/L/M/M** | **L/M/M/M** |
| 9 | M/M/M/M | M/M/M/M | M/M/M/M | **L/M/M/M** | **L/M/M/M** | M/M/M/M | M/M/M/M | M/M/M/M | **L/M/M/M** | M/M/M/M |
| all | M/M/M/M | M/M/M/M | M/M/M/M | M/M/M/M | M/M/M/M | M/M/M/M | M/M/M/M | M/M/M/M | M/M/M/M | M/M/M/M |
| Count | 0/0/0/0 | 6/3/2/0 | 1/0/3/0 | 2/0/0/0 | 3/0/3/0 | 0/0/0/0 | 6/4/0/1 | 1/1/1/0 | 1/1/1/0 | 1/0/2/0 |

We incorporated this regularization method into the GAN training process for the initial 80 epochs by adding the SD regularizer to the discriminator's loss computation for real image batches, with $\lambda = 0.8$ (Appendix D presents results for different $\lambda$ values). After 80 epochs we removed the regularizer for further training until 200 epochs, allowing the GAN image quality to improve.

The effect of SD is evident in Figure 3 , where the discriminator loss dynamics (illustrated by solid and dashed black lines) converge more closely during the SD application phase (up to epoch 80), suggesting increased discriminator invariance to RGF and thus mitigating the memorization problem. In addition, Tables 1 and 2 demonstrate that applying SD generally results in smaller z-scores, suggesting reduced memorization.

Finally, in Table 4 we used the p-values corresponding to the z-scores in Tables 1 and 2 (for MNIST data) to deduce whether the RGF is learned (L) or memorized (M). Note that all DM values are M, indicating a strong tendency of diffusion models to memorize RGFs. We observe that SD helps in mitigating memorization to some extent for GAN. For CompCars data, GAN with SD achieved learning in one case only (Table 3). We report results using two additional random seeds in Appendix F, further validating these findings.

## 7 Conclusion

We are interested in examining how generative models like VAEs, GANs and DMs learn rare generative factors (without explicit supervision). Through a systematic empirical study involving several generative factors and two datasets, we showed that generative models exhibit a propensity towards memorizing rare

generative factors. We demonstrated that regularization techniques such as spectral decoupling can mitigate this memorization tendency to a certain degree.

There are several intriguing directions for future research. Firstly, applying our framework to other types of generative models, to assess their efficacy in learning rare generative factors. Secondly, a deeper exploration into the learnability of rare generative factors across a broader array of (real-world) datasets would significantly enhance our understanding of how these models perform in diverse scenarios. Lastly, exploring the integration of novel regularization techniques (such as for VAE large $\beta$ values, normalizing flows Rezende & Mohamed (2015) and Generalized ELBO with Constrained Optimization Rezende & Viola (2018)) or architectural modifications could offer further insights into mitigating memorization and improving the learnability of rare generative factors.

While our current study explores how generative models learn rare generative factors (RGFs) from scratch in unsupervised settings, future work will investigate whether large pretrained models like Stable Diffusion can overcome the limitations we observed. This shift allows us to test if prior knowledge encoded during large-scale pretraining enables more balanced or robust learning of RGFs.

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
