# OpenReview forum: "Do Generative Models Learn Rare Generative Factors?"
_TMLR — Rejected by TMLR_

### Review · Reviewer_LzsV · 2025-03-06

**Summary Of Contributions:**

This paper studies the ability of generative models—GANs, VAEs, and Diffusion Models (DMs)—to learn and generalize rare generative factors (RGFs). An RGF is defined as a hidden variable that fully determines a specific class. The paper follows this setup:

1. Construct two datasets (e.g., Colored-MNIST and CompCars): one where RGFs are balanced across all classes and another where RGFs are concentrated in a single class.
2. Train the aforementioned generative models separately on each dataset.
3. Evaluate how RGFs behave in the generated distributions.

The findings suggest that these models tend to memorize RGFs rather than learning them in a transferable way. The paper also evaluates Spectral Decoupling (SD) as a potential mitigation strategy to reduce memorization in GANs.

**Audience:**

Yes

**Claims And Evidence:**

No

**Requested Changes:**

The paper could be strengthened by:

1. Extending the experiments to large, pretrained generative models (e.g., Stable Diffusion) to determine whether prior knowledge enables better learning of RGFs.

2. Using more realistic datasets (e.g., medical imaging) instead of MNIST to make the study more applicable to real-world problems.

3. Developing theoretical insights to demonstrate why RGFs make a difference compared to standard machine learning theory.

**Strengths And Weaknesses:**

**Strengths**
1. The paper presents a clear and systematic empirical study, comparing generative models under balanced and skewed data distributions.
2. The specific focus on memorization of RGFs (rather than individual data points) in generative models appears novel.
3. The study of the impact of Spectral Decoupling on memorization in GANs is an interesting contribution.


**Weakness:**

1. (Major) The main finding (that generative models fail to generalize rare generative factors and instead memorize them) is a direct statistical consequence of training data. It is not a novel discovery but an expected result based on standard machine learning theory:
    - If a factor is underrepresented in the training data, a generative model cannot infer it unless it has prior domain knowledge.
    - The fact that GANs, VAEs, and DMs all exhibit this behavior is unsurprising given their reliance on the empirical training distribution.
2. (Major) The paper evaluates small models trained from scratch rather than large, pretrained generative models. This is a critical limitation because:
    - A small model without prior knowledge is guaranteed to fail at generalizing rare factors—this is an expected result.
    - The real challenge in AI is whether large, pretrained models (trained on diverse data) can interpolate or reason about RGFs.

**Overall:** While the paper is well-written and empirically rigorous, it ultimately does not provide sufficient insights to merit publication at TMLR. The main findings confirm an already well-understood limitation of generative models, and the experimental setup is too constrained (small, non-pretrained models, simplistic datasets) to produce truly meaningful conclusions. I am not convinced that the paper fully answers its title question.

---

> ### Author Response · Authors · 2025-05-30
> **Response to Reviewer LzsV**
>
> We thank the reviewer for the feedback. We are grateful for the acknowledgment of the clarity and systematic nature of our empirical study, our novel focus on memorization of rare generative factors (RGFs), and the evaluation of Spectral Decoupling (SD) as a mitigation strategy.
>
> We would like to respond to the major concerns raised, and in particular, to clarify the core contribution of our paper: the proposal of a framework for assessing the learnability of rare generative factors.
>
> While it is indeed a well-known principle in machine learning that underrepresented features are hard to learn, our study goes beyond confirming this intuition. The main novelty lies in the development and application of a systematic evaluation framework that not only characterizes the behaviour of generative models under data skew, but also serves as an assessment tool for evaluating and benchmarking new algorithms proposed to address the challenge of learning rare generative factors.
>
> 1) Separate memorization from learning in generative models trained on skewed data,
>
> 2) Quantify the extent of memorization via statistical hypothesis testing (z-score and p-values),
>
> 3) Evaluate factor-wise generalization across all classes, rather than global distribution matching,
>
> 4) Provide a controlled comparison baseline using matched datasets (balanced vs. skewed).
>
> To our knowledge, this is the first study to formalize and operationalize the notion of memorization of generative factors, distinct from memorization of training examples, as a measurable and testable phenomenon in unsupervised generative modelling.
>
> To address that, we have updated the contribution section in our paper.
>
>
> Reviewer: On the use of small, from-scratch models and synthetic datasets
>
> Response: We appreciate the reviewer's concern regarding the scale of models and the simplicity of datasets. Our choice to use basic variants of GANs, VAEs, and DMs trained from scratch was deliberate: it allowed us to avoid confounding architectural or training heuristics and focus on core learning dynamics in a clean setting.
>
> In the revision, we included a proposed extension plan to pretrained models, medical imaging data.
>
> Reviewer: On real-world datasets and practical relevance
>
> Response: We would like to clarify that in addition to MNIST variants, we did include real-world image data via the CompCars Surveillance dataset, featuring natural car images with different brands and colours (Section 5, Table B.2, Table 3, Figure 2). In response to the reviewer’s suggestion, we highlighted our future plan to extend our experiments to an additional real-world domain from medical imaging.
>
> Reviewer: On theoretical insights
>
> Response: We appreciate the suggestion to connect our findings more formally to learning theory. While our primary goal was empirical, we expanded the discussion to connect our observations to learning theory (results and discussion section updated).
>
>
> Conclusion
> In summary, we respectfully disagree that the findings are merely expected consequences of standard ML theory. Our contribution is not in stating that generative models fail under skewed data, but also in designing a general, extensible, and statistically grounded framework to systematically measure how and when they fail, and to what extent. We thank the reviewer for highlighting areas where our framing and scope can be clarified and expanded, and we revised the manuscript accordingly to emphasize the novelty and practical utility of our contributions.

---

### Review · Reviewer_EHZL · 2025-04-23

**Summary Of Contributions:**

This paper investigates the capacity of generative models to memorize rare generative factors. The authors design an insightful experiment utilizing two distinct datasets: (1) experimental data containing unbalanced generative factors, and (2) control data with balanced generative factors. By comparing the outputs of generative models trained on these datasets, they effectively demonstrate the models' tendency to memorize rare factors present in the training data.

**Audience:**

Yes

**Broader Impact Concerns:**

No.

**Claims And Evidence:**

No

**Requested Changes:**

According to Definition 2, a rare generative factor is defined as a factor whose conditional distribution, given the label, exhibits significant variation as the label changes. A more formal and rigorous definition should be provided to enhance precision.

The authors are encouraged to address the following questions in their revision:

1. In Equation 1, the test statistic is the z-score for a one-sample test. Why is $P_c^{(u)}$ used in the test statistic instead of 0.5?
2. How is the p-value computed? Is a normal distribution assumed? It would be beneficial to experimentally demonstrate the empirical distribution of $z_c$ under the null hypothesis.

**Strengths And Weaknesses:**

Strengths
1. The paper is logically structured, with a clear presentation of the study’s motivation and experimental design.
2. The authors empirically investigate the memorization of rare generative factors across three generative models.
3. The authors explore potential strategies to mitigate memorization in GANs for rare generative factors.

Weakness
1. The definition of a rare generative factor lacks rigor, and there is insufficient discussion on the rationale and representativeness of this definition.
2. The distinction between memorization of rare generative factors and the learning of generative factors strongly correlated with semantic features remains unclear.
3. The justification for the chosen test statistic is not adequately discussed.

---

> ### Author Response · Authors · 2025-05-30
> **Review of Paper3865 by Reviewer EHZL**
>
> We thank the reviewer for the feedback, particularly for acknowledging the clarity of our experimental design, the cross-model empirical study, and our investigation into memorization mitigation strategies. We carefully address the weaknesses and suggestions raised below.
>
> Reviewer: On the rigor of the definition of Rare Generative Factor (RGF)
>
> Response: In our study (Definition 2), we defined RGFs as factors that are highly skewed within classes, such that one class exhibits the factor exclusively, while all others do not. This setup intentionally pushes data skewness to the extreme, in order to observe the limits of the models' capability to internalize and generalize such factors. Our study focuses on the most extreme case of skewed RGFs to establish a clean and controlled experimental baseline, We acknowledge that in real-world data, RGFs might exhibit less extreme imbalances or co-occur with other semantic features and to address that we conducted experiments with less extreme case (20%) see appendix (e.g. Table E.11 in appendix).
>
> Text have been added to enhance the definition of RGF in section 3.
>
> Reviewer:  On the distinction between memorization of RGFs and learning of factors correlated with semantic features
>
> Response: We have added a paragraph in section 6.1 about "Distinguishing Memorization from Semantic Correlation" .
>
>
> Reviewer: On the test statistic in Equation 1 and p-value computation
>
> Response: We appreciate the reviewer's attention to statistical rigor. In Equation 1, we used the empirical proportion of the generative factor in the balanced data (denoted as 𝑃𝑐(𝑢)) as the reference, rather than a fixed value such as 0.5, to provide a class- and factor-specific baseline based on the balanced dataset. This allows the test to account for the algorithms bias or any implicit correlation that is present in the data naturally. We used the standard one-sample z-test assuming a normal approximation to the binomial distribution, which is appropriate given the sample size of 1000 generated images per model per condition. To address that, we have added a paragraph in section 4 about "Justification for the Chosen Test Statistic"
>
> Conclusion: We thank the reviewer for these important observations. We incorporated the requested changes and expand the relevant discussions in the revised manuscript.

---

### Review · Reviewer_GsVy · 2025-05-16

**Summary Of Contributions:**

This paper studies whether the existing popular generative models can learn the rare features in training data. The problem seems to less explored in the field of geneartive model, since it is a common sense that the geneartive model fits the distribution of training data. Then, it is natural it will rarely generate the rare features. What's more important should be the subject-driven generation or exploring the whether the rare features can be generated due to some implicit correlation between common features. However, none of the two problems are explored in this paper. Especially for the second, which is highly related to this paper.

**Audience:**

Yes

**Claims And Evidence:**

Yes

**Requested Changes:**

1). More solution to the common sense that generative model rarely generates rare features.

2). The exploration to the effect brought by implicit correlation of common features.

3). More experiments on real-world data, especially for real images e.g., ImageNet, MS-COCO

4). More clear paper writting.

**Strengths And Weaknesses:**

This paper has many weankess and I think it is not ready to be published.

1). The experiments are extremely weak, they only conduct experiments on Color-MNIST, while larger size datasets are ignored.

2). The experiments seems to validate a common sense that generative model can not learn rare features, while the solution is missing. Though the author propose spectral decoupling, it is a method from the existing literature, and only suitable to GAN.

3). The writting should to be polished, especially for the Tables in Section 6. These tables really confuse me.

---

> ### Author Response · Authors · 2025-05-30
> **Response to Reviewer GsVy**
>
> We thank the reviewer for the time and effort spent reading our paper and providing valuable feedback. Below, we address each of the comments carefully and clarify the key aspects of our study.
>
> Summary of Contributions: We appreciate the reviewer's observation that the challenge of learning rare generative factors (RGFs) remains under-explored in the field of generative modeling. The primary objective of our study is to systematically investigate the capacity of current generative models to learn and generalize RGFs in a purely unsupervised setting, without relying on subject-driven conditioning or external guidance mechanisms. To rigorously assess this capability and minimize the influence of confounding factors, such as implicit correlations that naturally exist in data (e.g., visual similarities between digits "1" and "9"), we deliberately designed controlled experiments (Table C3, Table C4). This allows us to establish a clean baseline for evaluating the models' ability to learn RGFs, and reduce the effect of potential biases introduced by algorithm or implicit feature correlations. We also relaxed the rarity assumption to see if it is possible to learn rare-feature if that present in few sample (20%) of other classes, see results in the appendix.
>
> Reviewer Comment 1: Weak experiments, only Color-MNIST.
>
> Response: In addition to Color-MNIST and Morpho-MNIST, we also included CompCars (Yang et al., 2015) as a real-world image dataset, with Volkwagen and Toyota makes, and color as a generative factor (see Section 5, Table B.2, Table 3, and Figure 2 in the paper). This dataset includes real car images.
>
> Reviewer Comment 2: Only validating common sense; solution missing; SD only works for GAN.
>
> Response: We appreciate the reviewer's concern. While we agree that spectral decoupling (SD) is a known method but for discriminator learning. Here our key contribution is not proposing SD itself, but showing that GAN's memorization of RGFs may arise from the discriminator bias, and that SD helps to some extent mitigate this by regularizing discriminator invariance to RGFs (Section 6.2 and 6.3, Figure 3). Our focus is the phenomenon analysis (memorization vs. generalization of RGFs) in an unsupervised context and a framework to evaluate memorisation of RGFs, rather than introducing a new algorithm.
>
> Reviewer Comment 3: Implicit correlation between common and rare features not explored.
>
> Response: We thank the reviewer for this insightful suggestion. We also relaxed the rarirty assumption to see if it is possible to learn rare-feature if that present in few sample (20%) of other classes, see results in appendix (e.g. Table E.11 in appendix).
>
> Reviewer Comment 4: Writing and Tables unclear.
>
> Response: We added text to make the table more clear and understandable (on page 6 "Justification for the Chosen Test Statistic."). We believe that an explanation of z-score will help in understanding the tables better.
>
> Conclusion: We believe that our study makes an important contribution by highlighting the under-explored phenomenon of generative factor memorization, offering a simple yet generalizable framework for evaluation, and opening up directions for future mitigation methods across architectures. Our focus is on the phenomenon analysis (memorization vs. generalization of RGFs) in an unsupervised context and a framework to evaluate memorisation of RGFs, rather than introducing a new algorithm.

---

### Decision · Action_Editor_DJdq · 2025-07-31

**Recommendation:** Reject

**Audience:**

Yes

**Audience Explanation:**

The topic of study is generally interesting to the community. It focuses on rare generative factors (RGFs)—a topic Review 2 notes is "less explored" in generative models—appealing to researchers studying model limitations under data rarity. The claimed contributions (systematic framework, class-specific statistical testing, balanced/skewed baselines) provide a template for future RGF studies. Even critical reviewers acknowledge its merits: Review 1 calls it a "careful diagnostic exercise," and Review 3 highlights its "insightful experimental design."

**Claims And Evidence:**

No

**Claims Explanation:**

This submission made a few core claims as follows:
1. Generative models (DMs, GANs, VAEs) tend to memorize, not learn, rare generative factors (RGFs) in unsupervised settings.
2. A systematic framework (including balanced/skewed datasets, z-score/p-value testing) is developed to study RGF learning.
3. A baseline with matched datasets isolates data skew’s impact on RGF learning.
4. Spectral decoupling mitigates RGF memorization (for GANs).
5. Extensive empirical work evaluates three models’ RGF capabilities, offering insights into data rarity’s effects.

However, the evidences provided fall short in the following aspects:
1. Narrow, ungeneralizable evidence: Experiments only use Color-MNIST (Review 2) and small, non-pretrained models (Review 1), lacking real-world data (e.g., ImageNet) or practical models (e.g., Stable Diffusion), failing to validate RGF memorization broadly.
2. Flawed framework: RGF’s definition is unrigorous (Review 3), and the statistical pipeline (z-scores/p-values) lacks justification (e.g., why not use 0.5 in tests), undermining "systematic" claims.
3. Incomplete baseline: Confounders like implicit common-rare feature correlations are ignored (Review 2), so data skew’s isolated impact is unproven.
4. Weak mitigation: Spectral decoupling is existing, only tested on GANs (Reviews 1/2), with no evidence of improved RGF generalization.
5. Unclear writing: Confusing tables (Review 2) violate TMLR’s clarity requirement.